# Epiallelic variation of non-coding RNA genes and their phenotypic consequences

Jie Liu [1] & Xuehua Zhong [1] ✉

Epigenetic variations contribute greatly to the phenotypic plasticity and diversity. Current functional studies on epialleles have predominantly focused on protein-coding genes, leaving the epialleles of non-coding RNA (ncRNA) genes largely understudied. Here, we uncover abundant DNA methylation variations of ncRNA genes and their significant correlations with plant adaptation among 1001 natural *Arabidopsis* accessions. Through genome-wide association study (GWAS), we identify large numbers of methylation QTL (methylQTL) that are independent of known DNA methyltransferases and enriched in specific chromatin states. Proximal methylQTL closely located to ncRNA genes have a larger effect on DNA methylation than distal methylQTL. We ectopically tether a DNA methyltransferase MQ1v to miR157a by CRISPR-dCas9 and show de novo establishment of DNA methylation accompanied with decreased miR157a abundance and early flowering. These findings provide important insights into the genetic basis of epigenetic variations and highlight the contribution of epigenetic variations of ncRNA genes to plant phenotypes and diversity.

Epigenetic variants, referred to as epialleles with different degree of DNA methylation, are associated with a wide range of complex biological traits in plants and animals[1–3]. The classical mice epiallele, *agouti viable yellow* ($A^{vy}$), controls the variations of coat color with the fully methylated/silent allele showing brown coat, the unmethylated/active allele showing yellow coat, and partial methylated allele giving rise to mottling[4]. The first natural plant epiallele with clear molecular basis is the naturally occurred heritable methylation of *Lcyc* gene, which leads to the silence of *Lcyc* and change in the flower morphology of *Linaria vulgaris*[5]. Additionally, several other natural epialleles with functional importance have also been reported, such as *Colorless nonripening* (*Cnr*) locus for tomato ripening[6] and *Karma* for mantled fruit of oil palm[7]. Along with these well-characterized epialleles, a large number of epigenetic variations have been identified in natural and epigenetic recombinant inbred lines showing strong correlation with gene expression changes and phenotypic variations in *Arabidopsis*[1,2,8–13].

Current epiallele studies have mostly focused on the protein-coding genes, transposable elements (TEs), and differentially methylated regions on the genome[8,13–15]. Little is known about the epialleles of

ncRNA genes. The ncRNA genes have been viewed as "junk" for a long time but found to participate in many biological processes during recent decades[16,17]. Dysregulation of ncRNAs has been attributed to the cause of cancer and other human diseases. These disease-related ncRNAs has thus become drug targets[18–20]. Plant ncRNAs play important roles in many processes through the fine-tuning of gene expression at transcriptional and/or post-transcriptional level[17,21,22]. Studies have also revealed critical roles of ncRNAs in establishing and maintaining epigenetic states. For example, small interfering RNA (siRNA) is required in the RNA-directed DNA methylation pathway to establish de novo methylation and maintain asymmetric CHH (H is A, C, or T) methylation to ensure genome fidelity and stability[23,24]. Three long noncoding RNAs (lncRNAs), *COOLAIR*, *COLDAIR*, and *COLDWRAP*, are antisense/sense transcripts derived from *FLOWERING LOCUS C* (*FLC*) and are essential for vernalization through recruiting Polycomb repressive complex 2 (PRC2) to deposit H3K27me3 and epigenetically silence *FLC*[25–28]. Emerging evidence has shown that lncRNAs can physically interact with DNA methyltransferases in mammals and methylcytosine-binding protein VIM1 in plants to regulate DNA methylation[29,30]. In contrast to the well-studied role of ncRNAs in DNA

[1]Department of Biology, Washington University in St. Louis, St. Louis, MO 63130, USA. ✉e-mail: xuehuazhong@wustl.edu

methylation, the effect of DNA methylation on ncRNA function and regulation is largely unknown.

Compared to other types of ncRNAs, miRNAs have been well studied with important functions in gene expression and some developmental phenotypes[21]. Precise regulation of flowering time controlled by hundreds of genes is critical for plants to adapt to natural environments. Among these flowering genes, the miR156/157 family, which encodes microRNAs and represses *SQUAMOSA PROMOTER BINDING PROTEIN-LIKE* (*SPL*) genes, is a master regulator of vegetative phase change and confers age-related floral induction through the miR156/157-*SPL* module[31–34].

In natural populations, both genetic and epigenetic changes may contribute to the variation of complex diseases and phenotypes. Thus, it is challenging to dissect the epigenetic function from genetic contribution. While global epigenome perturbation by genetic mutational studies has revealed significant association of epigenetic marks with their inferred functions, understanding how epigenetic features at specific loci contribute to a given biological function remains largely unknown. Recently, CRISPR-based systems have been reported to manipulate the locus-specific methylation status of target genes[35–37] by either fusion of an inactive Cas9 (dCas9) with a prokaryotic DNA methyltransferase MQ1[35] or recruitment of NtDRMcd[36]/MQ1v[37] to target sites by a SunTag system. These epigenome-editing strategies can decouple the DNA methylation of a specific target locus from genetic mutations and/or methylation of other regions to directly investigate the importance of epigenetic variations.

Distinct from most studies on ncRNA regulation of chromatin functions, this study focuses on epigenetic regulation of ncRNA genes and the importance of epigenetic variations of ncRNA genes to plant phenotypes. We profile the methylation level of 5,099 ncRNA genes in *Arabidopsis* 1001 Epigenomes and find that 74% of them show large variations with coefficient of variations (CV) > 100%. We map large numbers of methylQTL through GWAS and find that local genetic variants are much more determinative than distal genetic variants, and that methyQTL are enriched in H3K9me2 regions. Furthermore, methylation level of ~1400 ncRNA genes are significantly correlated with plant adaptation-related traits. To directly test the methylation contribution to a given trait, we perform epigenetic editing of miR157a gene to demonstrate that DNA methylation change of a single ncRNA gene can make a dramatic alteration to plant adaptation phenotype, i.e., flowering time. This study highlights the widely existed epigenetic variations of ncRNA genes, their genetic basis, and their important contribution to plant phenotypes.

## Results

### Natural variations in DNA methylation of ncRNA genes

We selected 811 natural accessions (Supplementary Data 1) from both 1001 *Arabidopsis* Genomes[38] and Epigenomes[8] with the same growth conditions and profiled the methylation levels of ncRNA genes and protein-coding genes. We found that the methylation levels of ncRNA genes were lower in CG context and higher in non-CG (CHG and CHH) context compared to the protein-coding genes (Fig. 1a). This finding prompted us to focus on the diversity and importance of methylation of ncRNA genes.

According to the *Arabidopsis* Araport11 genome annotation, there are 5,099 ncRNA genes located on 5 chromosomes and divided into 8 groups. The number of ncRNA genes from each group varies from 78 (antisense RNA genes) to 2,444 (lncRNA genes), with lncRNA and antisense lncRNA genes making up the majority (68%, Fig. 1b). Methylation traits were determined by calculating the average methylation levels of each ncRNA gene in CG, CHG, and CHH context in 811 accessions. A total of 12,115 methylation traits remained after filtering with a missing rate of ≤10% (<81 accessions with missed value) and used for subsequent analysis. The CV of these 12,115 traits ranged from 3% to 876%, with an average of 188%, and 74% (8963) of them

were greater than 100% (Fig. 1c, Supplementary Data 2). For example, 86% (1700 of 1977) and 74% (136 of 183) of CG methylation traits of lncRNA and miRNA genes had CV greater than 100%, respectively (Fig. 1c, d). We also surveyed the average DNA methylation levels of each group in each accession and found that ncRNA genes had varied DNA methylation levels among different groups. Compared to TEs, ncRNA genes had much lower DNA methylation in all cytosine contexts. In non-CG context, most groups had higher methylation level than protein-coding genes. While in CG context, both sense lncRNA and antisense lncRNA genes had slightly lower DNA methylation comparing with protein-coding genes (protein-coding genes: $16.57 \pm 0.02$, lncRNA genes: $16.05 \pm 0.03$, antisense lncRNA genes: $12.28 \pm 0.02$, Mann-Whitney test $P < 2.00 \times 10^{-15}$, Fig. 1e). This observation explained the global higher CG methylation of protein-coding genes than that of ncRNA genes (Fig. 1a). Notably, miRNA genes showed much higher methylation levels in both CG and non-CG context (Fig. 1e).

To investigate the biological significance of DNA methylation of ncRNA genes, we first calculated the correlation coefficient between DNA methylation and latitude and longitude data of the natural accessions. We found that DNA methylation of 196 and 162 ncRNA genes was significantly correlated with latitude and longitude, respectively ($P < 6.08 \times 10^{-5}$, $N > 720$, Supplementary Data 3). Among these genes, 28 were significantly correlated with both latitude and longitude. To determine whether methylation of ncRNA genes was correlated with plant phenotypes, we gathered phenotype datasets (303 quantitative phenotypes) of natural *Arabidopsis* accessions from 12 studies[39] (Supplementary Data 4) and calculated the correlation between these phenotypes and DNA methylation of ncRNA genes. In total, 84% (254 of 303) of these phenotypes were significantly correlated with DNA methylation of 1468 ncRNA genes ($P < 6.08 \times 10^{-5}$, $N > 100$, Supplementary Data 4). After cataloging these phenotypes into 8 groups and constructing a network based on significant correlations with Cytoscape[40], we found that most (87%, 1801 of 2072) significant correlations fell into groups of fitness, flowering, plant structure, development, and root morphology (Fig. 1f). While for ion-, yield-, and metabolite-related phenotypes, far fewer correlations (13%, 271 of 2072) were detected (Fig. 1f). Together, these analyses revealed abundant variations of DNA methylation within ncRNA genes and high correlation of these variations with plant phenotypes.

### GWAS of DNA methylation of ncRNA genes

To determine the genetic basis of DNA methylation variations of ncRNA genes, we performed GWAS to map the methylQTL for CG, CHG, and CHH methylation of each ncRNA gene. The genotype data of 811 accessions extracted from 1001 *Arabidopsis* Genomes[38] contained 1,110,440 biallelic single nucleotide polymorphism (SNP) with minor allele frequency ≥5% and missing rate ≤25%. The association tests between SNPs and methylation traits were performed with EMMAX[41] using a mixed linear model with consideration of relatedness between accessions. To test this GWAS method, we ran GWAS on CHH methylation of CMT2 targeted TEs and detected signals around *CMT2* (Supplementary Fig. 1) consistent with published studies[8,15]. We then ran GWAS on 12,115 traits and detected 564,346 significant associations between 188,347 SNPs and methylation levels of 3,825 ncRNA genes after Bonferroni correction ($P \leq 9.01 \times 10^{-9}$, Fig. 2a). Surprisingly, very few significant SNPs (218 of 188,347) in genes (including 2 kb upstream and 2 kb downstream region) were reported to regulate DNA methylation (Fig. 2a). In *Arabidopsis* Col-0 accession, DNA methylation is maintained by DNA methyltransferases MET1 for CG, CMT3 for CHG, CMT2 and DRM2 for CHH context[24,42,43]. In our GWAS results, only 73 significant SNPs were in genes encoding these methyltransferases. When comparing methylation levels of 811 accessions with *met1*, *cmt3*, *cmt2*, and *drm2* mutants, we found that 100%, 99%, 25%, and 13% of these 811 accessions had higher methylation levels than the

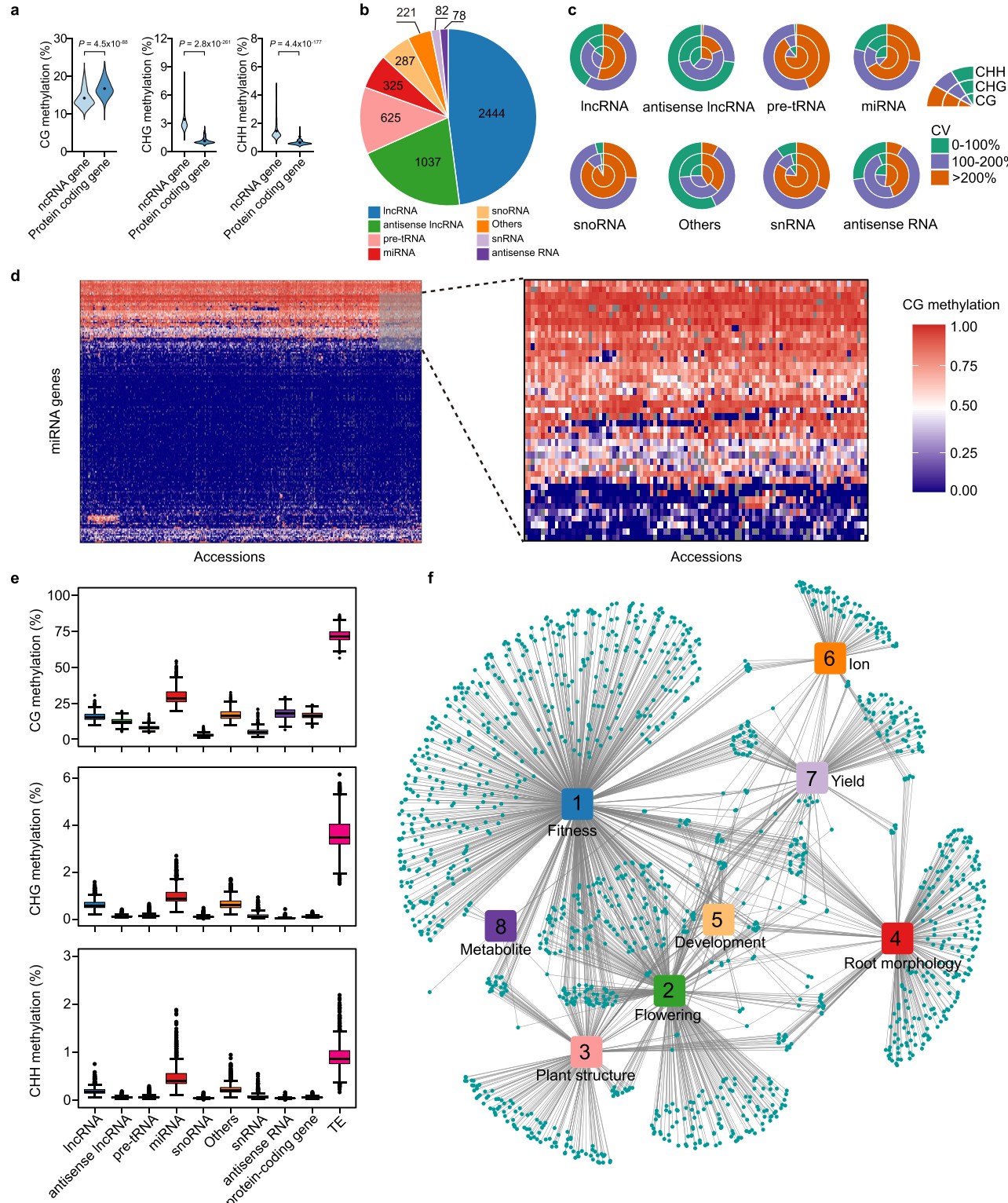

**Fig. 1 | Natural variation of DNA methylation of ncRNA genes. a** DNA methylation of ncRNA genes and protein-coding genes in CG, CHG, and CHH context among 811 *Arabidopsis* accessions. The *P*-value was calculated by Mann−Whitney test. **b** The classification and number of ncRNA genes according to Araport11 genome annotation of *Arabidopsis*. **c** The variation of DNA methylation measured with coefficient of variation (CV). The colors indicate the degree of variation with three layers corresponding to CG, CHG, CHH methylation, respectively. **d** The heatmap of CG methylation of miRNA genes. Right panel is the zoom-in view of the gray-boxed region in the left panel. **e** The distribution of average CG, CHG, CHH methylation of each group in 811 accessions. The horizontal lines within the boxes represent the median; the dots represent outliers (more than 1.5 × IQR from the hinge, where IQR is distance between the first and third quartiles); and the lower and upper hinges of the box represent the 25th and 75th percentiles, respectively. *N* = 811 for each boxplot. **f** The network of significant correlations between methylation of ncRNA genes and plant phenotypes. Each dot indicates one ncRNA gene and each line indicates significant correlation between the DNA methylation of this gene and the plant phenotype. Source data are provided as a Source Data file.

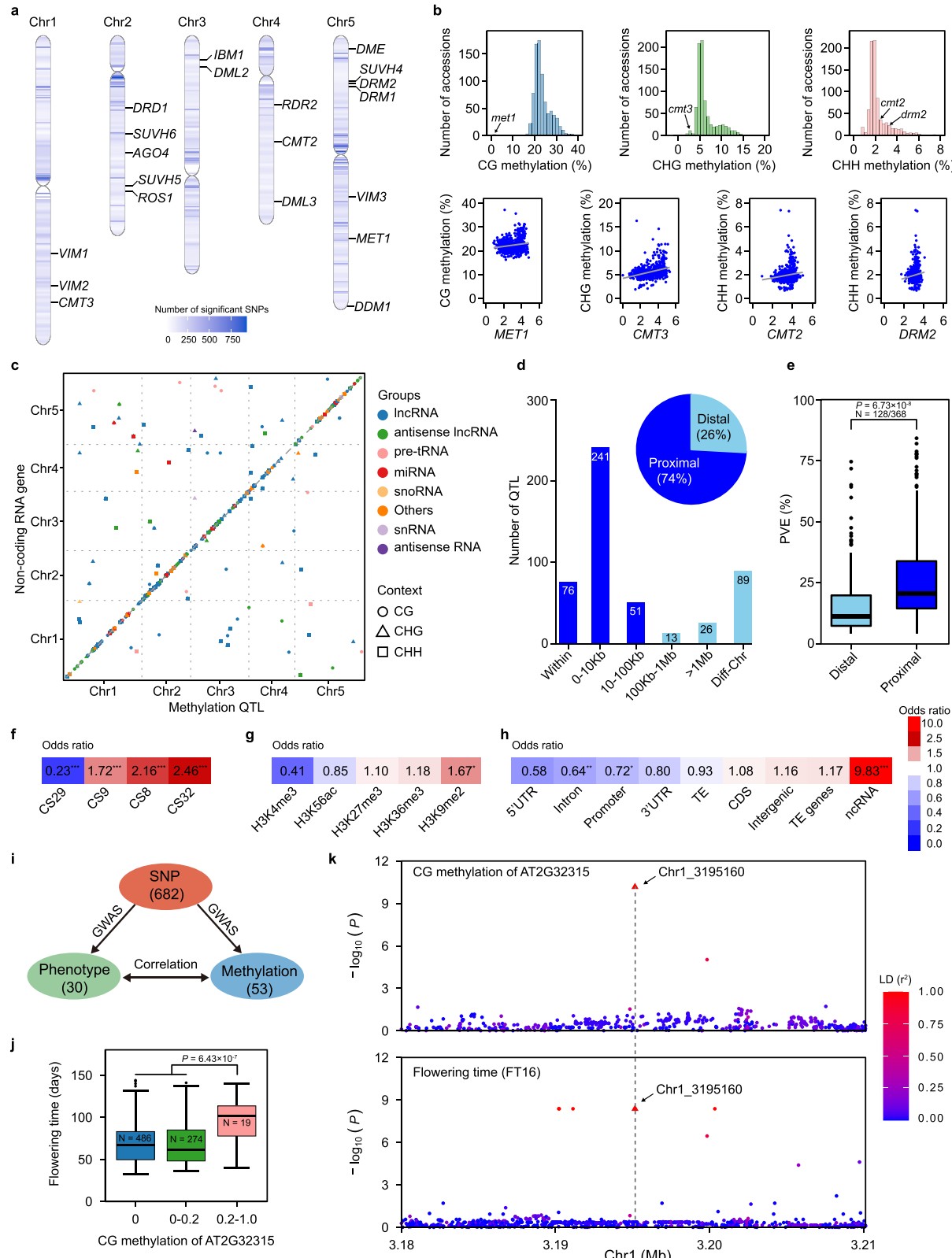

corresponding mutants (Fig. 2b). This indicates that these natural accessions were much more tolerant with CMT2/DRM2-mediated CHH methylation loss than MET1- and CMT3-mediated CG and CHG methylation loss, respectively. We did not observe any significant correlations between expression of these methyltransferases with corresponding CG/non-CG methylations (Fig. 2b). Taken together, these results suggest that the genetic basis of methylation of ncRNA

genes were complex and that known methyltransferases were not responsible for the observed DNA methylation variations of ncRNA genes in the natural population.

To explore the common features of methylQTL and exclude associations caused by linkage disequilibrium, we applied multiple stringent filtration steps (Methods) and kept 496 high-confidence methylQTL for subsequent analysis (Fig. 2c and Supplementary

**Fig. 2 | GWAS of DNA methylation of ncRNA genes. a** The distribution of identified significant SNPs and some known methylation related genes on *Arabidopsis* genome. The numbers of significant SNPs are calculated in each 100Kb window. **b** The histogram of CG, CHG. CHH methylation levels among 811 accessions and *met1, cmt3, cmt2, drm2* mutant (top) and correlation plots of expression (log2(TPM + 1)) and methylation (bottom). **c** The distribution of 496 high confidence methylQTL. The y-axis of each dot indicates the physical position of ncRNA gene, and the x-axis is the position of corresponding methylQTL. **d** The number of methylQTL divided by the distance of associated SNP to the corresponding ncRNA gene. The pie chart shows the proportion of proximal (<100Kb) and distal (>100Kb) methylQTL. **e** Boxplot of phenotypic variation explained (PVE) by methylQTL. The horizontal lines within the boxes represent the median; the dots represent outliers (more than 1.5 * IQR from the hinge, where IQR is distance between the first and third quartiles); and the lower and upper hinges of the box

represent the 25th and 75th percentiles, respectively. The *P*-value is calculated by one-way ANOVA analysis. **f–h** Enrichment analysis of methylQTL. The odds ratio and significance are calculated with Fisher's exact test. ***$P < 0.001$; **$P < 0.01$; *$P < 0.05$. **i** The SNPs simultaneously identified by GWAS of correlated phenotype and methylation of ncRNA gene. **j** The CG methylation of AT2G32315 is significantly correlated with flowering time. The horizontal lines within the boxes represent the median; the dots represent outliers (more than 1.5 × IQR from the hinge, where IQR is distance between the first and third quartiles); and the lower and upper hinges of the box represent the 25th and 75th percentiles, respectively. The *P*-value is calculated by one-way ANOVA analysis. **k** SNP Chr1_3195160 is significant in GWAS of CG methylation of AT2G32315 (top) and flowering time (bottom). The color of each dot indicates linkage disequilibrium level between Chr1_3195160 and the SNP. Source data are provided as a Source Data file.

Data 5). Using the distances between representative SNPs of each methylQTL and the corresponding ncRNA genes, we classified them as proximal (<100Kb) or distal (>100Kb) methylQTL and found that most methylQTL (74%, 368 of 496) were proximal methylQTL (Fig. 2c, d). Using AT4G05535 that encodes a long noncoding RNA as an example, GWAS of its CG methylation identified a single significant signal with representative SNP located in its intron region ($P = 1.71 \times 10^{-107}$, Supplementary Fig. 2) and the phenotypic variation explained (PVE) by this methylQTL was 57%. Most of the distal methylQTL (70%, 89 of 128) were located on different chromosomes of the corresponding ncRNA genes (Fig. 2c, d). When comparing the PVE of proximal and distal methylQTL, we found that proximal methylQTL has greater effect than the distal methylQTL (26% vs. 17% on average, $P = 6.73 \times 10^{-8}$, Fig. 2e). DNA methylation of 4 ncRNA genes were simultaneously controlled by proximal and distal methylQTL (Supplementary Data 5). The CHG methylation of AT2G06395, which is located on chromosome 2 and encodes a long noncoding RNA, was regulated by one proximal methylQTL (representative SNP located in 398 bp downstream, $P = 2.23 \times 10^{-26}$, Supplementary Fig. 3, bottom panel) and another distal methylQTL (on chromosome 1, $P = 4.86 \times 10^{-13}$, Supplementary Fig. 3, top panel). The PVE of the proximal methylQTL was much higher than that of the distal methylQTL (18% vs. 7%) for the CHG methylation of AT2G06395. These results demonstrated the enrichment and the higher PVE of proximal genetic variants in regulating DNA methylation variations of ncRNA genes.

Further analysis of representative SNPs against chromatin states defined by two independent studies[44,45] revealed significant enrichments of these SNPs in repressive chromatin states CS8, CS9, CS32 marked with H3K9me2, and depletions in chromatin state CS29 colocalized with the promoter region (Fig. 2f, g). We carried out enrichment analysis against different genomic features and found that these representative SNPs were significantly enriched in the ncRNA gene regions and depleted in promoter and intron regions (Fig. 2h). This is consistent with the idea that proximal genetic variants play important roles in regulating DNA methylation of ncRNA genes.

Given the strong correlations of ncRNA gene methylation and phenotypes, we determined whether significant SNPs for ncRNA methylation were also presented in GWAS of the corresponding correlated phenotypes by obtaining significant SNPs from publicly available studies[46] and searching for the overlaps. 682 SNPs were simultaneously detected in GWAS results of correlated pairs of 30 phenotypes and methylation of 53 ncRNA genes (Fig. 2i). The CG methylation of AT2G32315 was significantly correlated with flowering time at 16 °C with higher methylation accessions (>20% of CG methylation) showing late flowering (Fig. 2j). Similarly, SNP Chr1_3195160 was significantly associated with both CG methylation of AT2G32315 and flowering time (Fig. 2k). AT2G32315 encodes an antisense long noncoding RNA that overlaps with AT2G32310 encoding a CCT domain protein. CCT genes are conserved classical flowering genes among

many plant species, such as *CO* in *Arabidopsis*[47], *ZmCCT* in maize[48,49], and *Ghd7* in rice[50]. Together, these results showed that methylation variations of ncRNA genes were highly correlated with the flowering and adaptation-related phenotypes.

### miR157a abundance is negatively correlated with flowering time

To directly test the impact of DNA methylation on ncRNA function, we focused on miR157a as a case study because there was a significant methylQTL locating ~800 bp downstream ($P = 1.15 \times 10^{-30}$ for CHG, Fig. 3a; $P = 3.13 \times 10^{-25}$ and $1.43 \times 10^{-23}$ for CG and CHH, Supplementary Fig. 4). The correlation analysis showed that DNA methylation of miR157a had positive correlation with longitude (Pearson's r = 0.15, N = 768, $P = 3.51 \times 10^{-5}$, Supplementary Data 3). The miR156/miR157 family redundantly regulated flowering time and knockout of miR157a in miR156 mutant had much earlier flowering than the single mutant[31–34]. Knockout of miR157a alone had no obvious change due to the redundancy of miR156/miR157 family[32]. However, it is unclear whether knock-down or altered abundance of miR157a has any impact on flowering. Since DNA methylation regulates gene expression, we first investigated the relationship between the abundance of miR157a and flowering time. To avoid the potential redundancy issue, we generated knockdown and overexpression transgenic lines through constitutively expressing a miR157a-specific short tandem target mimic (STTM[51]) and miR157a coding sequence driving by the CaMV35S promoter, respectively. We measured the abundance of mature miR157a with stem-loop primers[52] and found that mature miR157a abundance was significantly reduced in the two independent knockdown lines and increased in overexpression lines (Fig. 3b). We found that the knockdown and overexpression plants demonstrated early and late flowering phenotype, respectively (Fig. 3c–e). Previous studies showed that miR156/miR157-*SPL* module, in which miR156/miR157 repressed the expression/translation of *SPL* genes, was a key flowering pathway[33,34,53]. Consistently, we noted that nine *SPL* genes were significantly upregulated in miR157a knockdown lines (Fig. 3f) and downregulated in the overexpression lines (Fig. 3g).

To further explore the relationship between abundance of miR157a and flowering time, we obtained six independent transgenic lines with enhanced miR157a expression driven by its native promoter. Compared with WT, all six transgenic lines showed increased abundance of miR157a (Fig. 3h) and late flowering phenotype (Fig. 3i, j and Supplementary Fig. 5). Furthermore, there was a strong positive correlation between miR157a level and flowering time among these 6 transgenic lines (Pearson's r = 0.82, $P < 0.05$, Supplementary Fig. 6), suggesting that miR157a negatively regulates flowering time likely through repressing *SPL* expression.

### Tethering MQ1v to miR157a gene induces DNA methylation and early flowering

To directly test the role of DNA methylation in regulating miR157a expression, we utilized a SunTag-MQ1v system with an enhanced

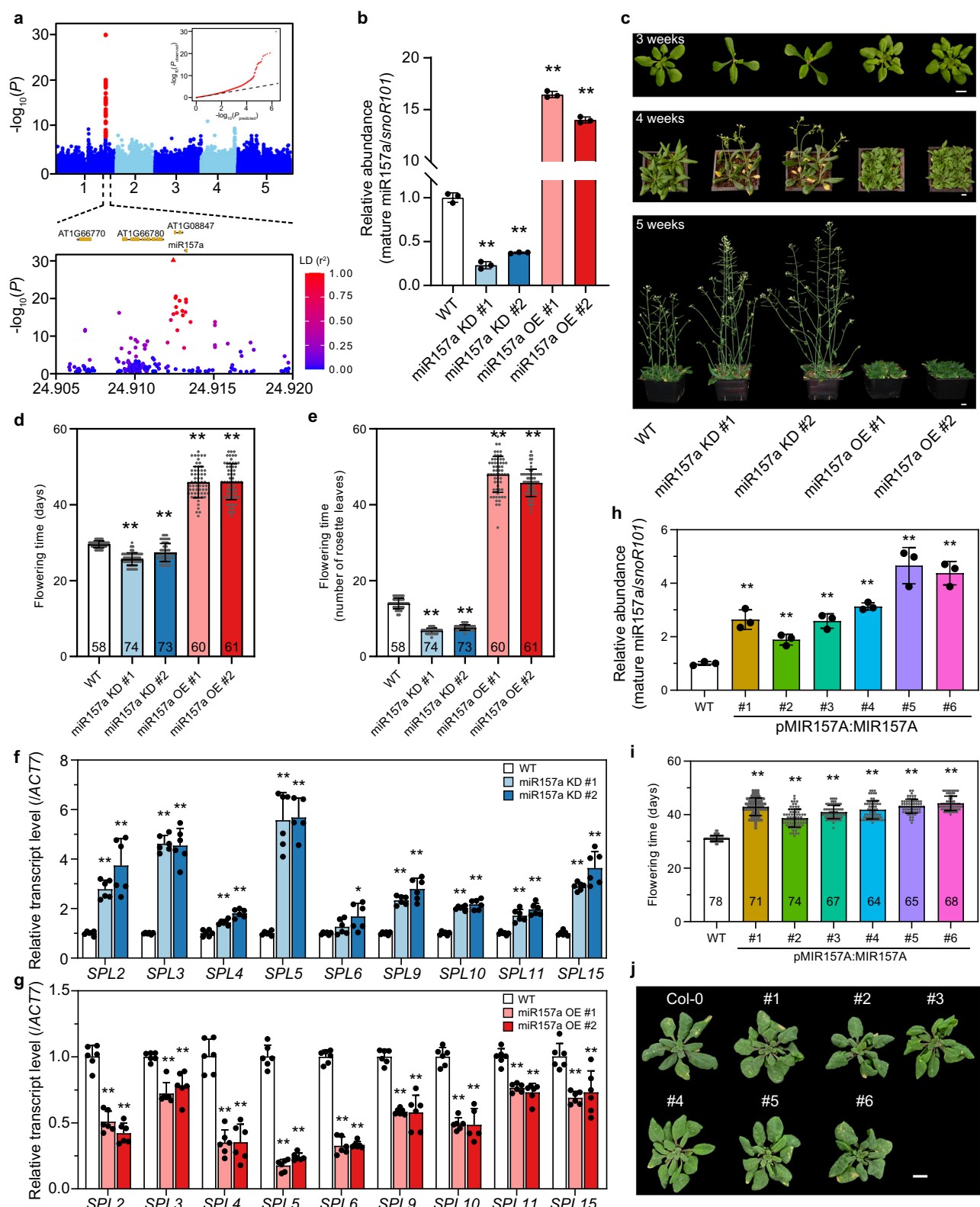

prokaryotic CG DNA methyltransferase MQ1v[35,37] to de novo methylate and generate an epiallele of miR157a in Col-0 background. In this system, miR157a specific sgRNA guided dCas9 to its target, which then recruited MQ1v to methylate nearby CG dinucleotides (Fig. 4a). We found an approximately 50% increase in CG DNA methylation at miR157a gene in SunTag-MQ1v line compared to WT (Fig. 4b). As expected, CHG and CHH methylation had no significant changes

(Supplementary Fig. 7), suggesting that MQ1v can specifically methylate CG dinucleotides at miR157a.

Compared to WT, we noted an early flower phenotype in SunTag-MQ1v transgenic plant (number of days to flowering: 25 vs. 30 in WT, $P = 4.87 \times 10^{-59}$; leaf number: 10 vs. 14 in WT, $P = 6.45 \times 10^{-75}$, Fig. 4c–e). Next, we measured the abundance of mature miR157a and found only 26% mature miR157a in SunTag-MQ1v compared to the WT

**Fig. 3 | miR157a regulates flowering time. a** GWAS of CHG methylation of miR157a gene. The inset panel is quantile-quantile plot of *P*-values. The representative SNP for this peak is indicated by a red triangle (lower panel). The color of each dot represents the linkage disequilibrium level with the representative SNP. **b** The relative abundance of mature miR157a in two independent knockdown and overexpression transgenic lines. For each measurement, three replicates are used. The *P*-value is calculated by one-way ANOVA analysis. Data are presented as mean ± SD. **\*\*P < 0.01. c** The flowering phenotype of WT and transgenic lines at three time points. bar, 1 cm. **d**, **e** The quantification of flowering time by days (**d**) and rosette leaf numbers (**e**). The number in each bar is the number of plants used for phenotyping. The *P*-value is calculated by one-way ANOVA analysis. Data are presented as mean ± SD. **\*\*P < 0.01. f**, **g** The relative expression level of *SPL* genes in knockdown (**f**) and overexpression transgenic lines (**g**). The *P*-value is calculated by one-way ANOVA analysis. Data are presented as mean ± SD. \*\*P < 0.01; \*P < 0.05. **h** The abundance of mature miR157a in WT and six independent *pMIR157A:MIR157A* transgenic lines. The *P*-value is calculated by one-way ANOVA analysis. Data are presented as mean ± SD. **\*\*P < 0.01. i**, **j** Flowering phenotype of WT and *pMIR157-A:MIR157A* transgenic lines. The number in each bar (**i**) is the number of plants used for phenotyping. The *P*-value is calculated by one-way ANOVA analysis. Data are presented as mean ± SD. \*\*P < 0.01. Source data are provided as a Source Data file.

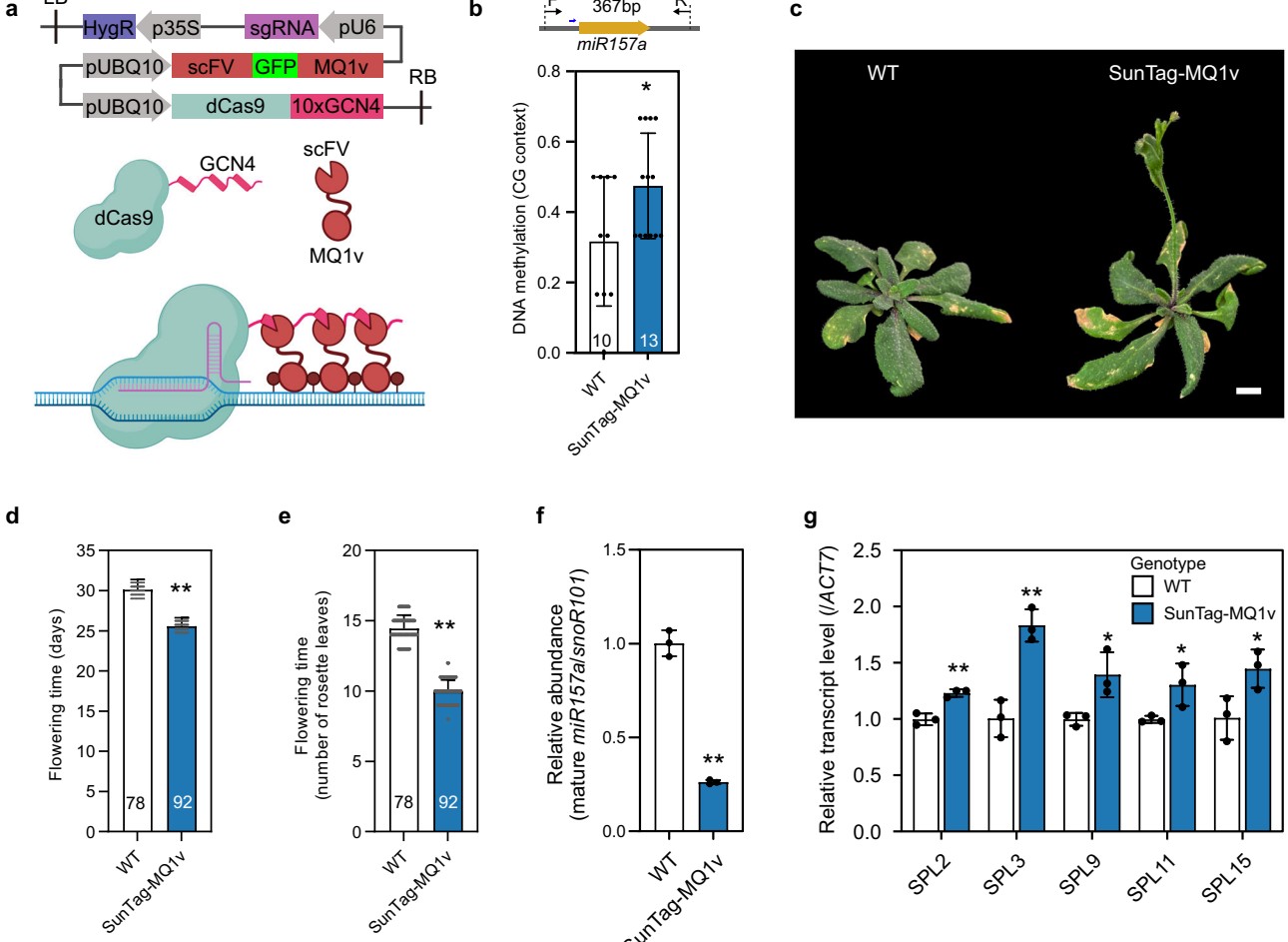

**Fig. 4 | Tethering MQ1v to miR157a gene induces DNA methylation and early flowering. a** The schematic demonstration of SunTag-MQ1v system used for de novo methylation of miR157a gene. It was created with BioRender.com **b** Measurement of DNA methylation levels in WT and SunTag-MQ1v transgenic line. The upper panel shows the location of miR157a and the primer used for BS-PCR. The blue arrow shows the location of sgRNA. The number in each bar shows the number of sequenced clones. The *P*-value is calculated by one-way ANOVA analysis. Data are presented as mean ± SD. \*P < 0.05. **c**–**e** The flowering phenotype of WT and SunTag-MQ1v line. The number in each bar (**d**, **e**) indicates the number of plants used for measuring flowering. The *P*-value is calculated by one-way ANOVA analysis. Data are presented as mean ± SD. **\*\*P < 0.01. f** Quantification of mature miR157a in WT and SunTag-MQ1v line. For each measurement, three replicates are used. The *P*-value is calculated by one-way ANOVA analysis. Data are presented as mean ± SD. **\*\*P < 0.01. g** Expression levels of *SPL* genes measured with RT-qPCR. For each measurement, three replicates are used. The *P*-value is calculated by one-way ANOVA analysis. Data are presented as mean ± SD. \*\*P < 0.01; \*P < 0.05. Source data are provided as a Source Data file.

($P = 5.31 \times 10^{-5}$, Fig. 4f). This negative correlation was consistent with the relationship observed in knockdown lines (Fig. 3b). We further measured the expression levels of *SPL* genes and found a significant increase of *SPL* transcript levels in SunTag-MQ1v line (Fig. 4g, Supplementary Fig. 8). Taken together, this MQ1v targeting system provided direct evidence that DNA methylation of miR157a could lead to its decreased expression, which then derepressed *SPL* genes and induced early flowering.

## Discussion

The widespread distribution of significant SNPs association with methylation of ncRNA genes across the genome indicated the complexity of the genetic basis. Our GWAS results did not identify any common proximal/distal methylation QTL, distinct from the individual level in which DNA methylation was controlled by a limited number of methyltransferases. At the population level, CG, CHG, and CHH methylation variations were independent of *MET1, CMT3*, and *CMT2/*

*DRM2*, respectively. The ~500 high confidence methylQTL identified in this study were highly enriched in certain specific chromatin states, indicating the existence of some common features underlying these methylQTL. Several studies highlighted the tight link between DNA methylation variations and 3D genome organization. Half of variable CTCF (a key regulator of the 3D genome structure) binding sites among 19 different human cell types were reported to be linked to differential DNA methylation[54]. The causal function of prostate cancer risk-associated SNP identified with GWAS could be disrupted by CTCF deposition, which is DNA methylation dependent[55]. When expressing DNA methyltransferases in *Saccharomyces cerevisiae*, Buitrago et al. observed globally increased *cis*-contacts and decreased *trans*-contacts[56]. In *Arabidopsis*, DNA methylation and other repressive epigenetic marks were intensely colocalized with the highest levels of chromatin interactions, and both *met1* and *ddm1* mutants showed easily detectable chromatin interaction changes compared with wild type[57,58]. Thus, 3D genome organization might be an important factor of epigenetic diversity.

In our GWAS results, the majority of methylation QTLs was proximal and had larger effect in controlling methylation variation than the distal QTLs. There were multiple studies in which humans exhibited similar patterns where methylation QTL associations predominantly occurred in *cis*[59,60]. 15% significant SNPs (29,118 of 188,347) were distributed in genomic regions that were neither TEs nor encoded protein or any annotated ncRNAs (Supplementary Data 6). These notable observations suggest that some genomic sequences might affect DNA methylation of nearby sites. In line with this hypothesis, previous studies in maize and *Arabidopsis* have shown that insertion of some specific retrotransposon families/siRNA-targeted TEs could lead to the spreading of DNA methylation to adjacent regions[61–64]. When depositing centromere repeat sequences into the intron region of *ABI5* in *Arabidopsis ibm1* mutant background, it could induce the establishment and spreading of DNA methylation[65]. Gene duplication, a tail-to-tail inverted repeat present in *Arabidopsis PAI* genes, was reported to cause dense methylation, while *PAI* in the accessions lacking this gene duplication was not methylated[66]. Despite these well-established causal genetic variants, more and more studies have revealed the genome wide correlations between structure variations (SVs) and DNA methylation. In *Arabidopsis*, 22%-50% of SVs identified through aligning the physical genome maps of 8 accessions to TAIR10 reference were hypo-/hyper- methylated[8]. One specific type of SVs, insertions enriched with DNA type TEs, was found to have much higher DNA methylation than flanking sequences in maize[67]. Somatic SVs were associated with changes of nearby DNA methylation and gene expressions through analysis of integrated data from 1400 human cancers[68]. Copy number variations, another type of SV, were also found to be significantly associated with CG methylation[69]. These preliminary explorations shed light on the potential important role of SV in regulating DNA methylation.

In this study, we found that epigenetic variations of ncRNA genes were highly correlated with plant adaptation phenotypes, and generation of an epiallele of miR157a could shape plant flowering time. About ~1400 ncRNA genes whose DNA methylation were significantly correlated with plant adaptation phenotypes including latitude, longitude, fitness, flowering, plant structure, root morphology and development. Most of these ncRNA genes have unknown function thus far. Regardless, we were able to find some genes with clear functions, such as miR157a, some miRNAs targeting transcription factors, and *COOLAIR* (AT5G01675) whose methylations were significantly correlated with longitude, flowering time, and other plant adaptation phenotypes in this study (Supplementary Data 3, 4). *COOLAIR* encodes an antisense long noncoding RNA and mediates cold induced silencing of *FLC*, a key determinant of flowering time[25,26]. Epigenetic editing of miR157a successfully de novo methylated CG at target sites and decoupled the genetic variations with DNA methylation change. We provided the

direct evidence that methylation change of miR157a, which was independent of genetic variation, led to early flowering and provided a strong case demonstrating that epigenetic change could make an enormous contribution to plant phenotypes. Besides miR157a, we uncovered a large number of correlations suggesting that epigenetic variations of ncRNA genes might be much more important and may have been underestimated in shaping plant phenotype and adaptation.

## Methods

### Plants materials
The Columbia-0 (Col-0) ecotype was used as the background for all transgenic lines. Seeds were germinated on 1/2 MS plates after 3 days cold treatment (4 °C) and then harvested for experiments or transferred to soil at 7–10 days seedling stage. Seedlings and plants were grown under long-day conditions (16 h light/8 h dark) at 22 °C.

### Vector construction and plant transformation
For knockdown of miR157a, STTM157a was synthesized by GenScript (Piscataway, NJ) and then cloned into pCAMBIA1300-p35S. For overexpression of miR157a, 362 bp covering 175 bp upstream, 98 bp coding sequence, and 89 bp downstream of miR157a was amplified using gDNA as a template and cloned into pCAMBIA1300-p35S. For transgene of *pMIR157A:MIR157A*, 2130 bp covering 1850bp upstream, 98 bp coding sequence, and 182 bp downstream of miR157a was amplified using gDNA as template and cloned into pCAMBIA1300. For SunTag-MQ1v, dCas9-MQ1(Q147L) was amplified from pLV hUbC-dCas9-MQ1(Q147L)-EGFP (Addgene: plasmid #89793), and then cloned into BsiWI digested pEG302 22aa SunTag NtDRMcd nog (Addgene: plasmid #115488) to replace NtDRMcd. All constructs were transferred into *Agrobacterium tumefaciens* strain GV3101 by electroporation and introduced into *Arabidopsis* by the floral dipping method. The primers used for transgenic vector construction are listed in Supplementary Data 7.

### Population methylomes and GWAS
BEDtools v2.29.0[70] was used to calculate methylation levels from the 1001 *Arabidopsis* Methylomes[8] (NCBI GEO accession: GSE43857). To avoid the impact of other factors such as temperature, we only used data from 811 accessions. When calculating DNA methylation, only cytosine sites with coverage greater than 4 were adopted. The genotype data consisting of 1,110,440 SNPs of 811 accessions were extracted from 1001 *Arabidopsis* Genomes[38]. LD decay was calculated with PLINK v1.9[71] with parameters "--file genotype_file --r2 --ld-window-kb 32000 --ld-window 1000 --ld-window-r2 0". The distance and $r^2$ value were then extracted from output file and $r^2$ value was averaged based on different distances. The LD decay was plotted with R software v3.6.1 (https://www.r-project.org/) using data with distance ≤100Kb. Genotype data was then filtered with PLINK v1.9[71] (parameter: --indep-pairwise 50 10 0.1) and 46,056 SNPs were kept and used for kinship matrix calculation with EMMAX[41]. GWAS was performed with EMMAX[41] using default parameter (emmax -v -d 10 -t [genotype] -p [phenotype] -k [kinship]).

### Filter of GWAS results and enrichment analysis
SNPs with $P \le 9.01 \times 10^{-9}$ (Bonferroni correction, 0.01/n where n is the number of SNPs used in GWAS) were kept for further analysis. We calculated the linkage disequilibrium (LD) decay ($r^2 \ge 0.1$ within 8Kb, Supplementary Fig. 9) and SNP density (~150 SNPs/16Kb). This LD decay distance was very similar to previously reported 10Kb which was estimated with 19 *Arabidopsis* accessions[72]. For a given significant SNP, it was preserved only if the number of other significant SNPs within nearby 16Kb was greater than 20 (Supplementary Fig. 10a, b). Based on the LD decay, SNPs were grouped into clusters if the distance of two consecutive SNPs was ≤8Kb. Only GWAS results for traits with cluster number no more than 20 were retained (Supplementary Fig. 10c, d).

The most significant SNP in each cluster was regarded as representative SNP and clusters whose representative SNP was in LD ($r^2 \geq 0.1$) with more significant representative SNP in other clusters for same trait were excluded. For the enrichment analysis of significant representative SNPs, Fisher's exact test was used to calculate the $P$-value and odds ratio with the distribution of 1,110,440 SNP as control. The chromatin states data were obtained from two studies[44,45]. The H3K9me2 peaks were reported by Zhang et al[73]. and other H3 histone marks were reported by Bewick et al[74]. The calculation and adjustment of $P$-value were done with R software (https://www.r-project.org).

### RNA extraction and expression quantification

For quantification of mature miR157a, small RNAs were extracted from 10 days old seedlings following the protocol reported by Zhang et al[52]. Briefly, we homogenized ~100 mg leaves and added 1 mL TRIzol (Invitrogen) and 0.4 mL water. The homogenate was stored at room temperature for 10 min after mixing well and then centrifuged at 12,000 rpm for 15 min at 4 °C. The supernatant was transferred to a new tube and added 0.8 volume of isopropanol (Thermo Scientific™). The mixture was stored at 4 °C for 30 min after mixing well and then centrifuged at 12,000 rpm for 15 min at 4 °C. After discarding the supernatant, the pellet in the tubes was washed twice with 70% isopropanol (Thermo Scientific™). The pellet was then dissolved with 40 μL RNase-free water and used for following experiments. For quantification of *SPL* genes, total RNAs were extracted from 10-day-old seedlings using the TRIzol reagent (Invitrogen). The RNAs were then treated with DNase I (New England Biolabs) and reverse transcribed with ProtoScript® II Reverse Transcriptase (New England Biolabs). Quantitative PCR of each sample with three replicates was performed using SYBR Green qPCR Master Mix (Vazyme) and a CFX96 Real-Time System (Bio-Rad). The relative transcript/miRNA abundance was calculated and normalized to that of Col-0 using the $2^{-\Delta\Delta Ct}$ method with *ACT7/snoR101* as the internal control. The primers used for RT-qPCR are listed in Supplementary Data 7.

### Quantification of DNA methylation by bisulfite method

For measuring DNA methylation of the target site, we used bisulfite-PCR sequencing (BS-PCR). Genomic DNA was digested with RNase A and then treated with bisulfite (EZ DNA Methylation-Lightning™ Kit). Eluted gDNA was used as template to run PCR with KAPA HiFi HotStart Uracil+ ReadyMix Kit. PCR products were purified (BIOLINE, ISOLATE II PCR, and Gel Kit), ligated to pCR-Blunt vector (Zero Blunt® PCR Cloning Kit, Thermal Fisher Scientific), and then Sanger sequenced. For each sample, at least ten clones were sequenced. The primers used for bisulfite PCR are listed in Supplementary Data 7.

### Statistical analysis

Statistical analyses were performed with Excel for one-way ANOVA analysis, R for Fisher's exact test and Mann-Whitney test, EMMAX[41] for GWAS $P$-values. Data are presented as mean ± SD as indicated. For the correlation analysis (Supplementary Data 3–6), the "corr.test" function in R "psych" package was used.

### Reporting summary

Further information on research design is available in the Nature Portfolio Reporting Summary linked to this article.

## Data availability

All data generated or analyzed during this study are included in this published article. Source data are provided with this paper. The 1001 *Arabidopsis* Methylomes can be obtained under NCBI GEO accession: GSE43857. The genotypic data of 1001 *Arabidopsis* Genomes can be obtained in 1001 Genomes Data Center https://1001genomes.org/data/GMI-MPI/releases/v3.1/. Source data are provided with this paper.

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

## Acknowledgements
We thank Xinrui Ji for screening homozygous transgenic lines of miR157a, Taiji Kawakatsu for providing the list of CMT2 targeted TEs, and Maddy Frank for manuscript edits. This work was supported by grant from NSF (MCB-2043544) and NIH (R35GM124806).

## Author contributions
J.L. performed data analysis and conducted all experiments. J.L. and X.Z. designed this study and prepared the manuscript.

## Competing interests
The authors declare no competing interests.
