## [Peer Review File · Nature Communications]

Epiallelic variation of non-coding RNA genes and their phenotypic consequencesEditorial Note: Parts of this Peer Review File have been redacted as indicated to remove third-party material where no permission to publish could be obtained.

REVIEWER COMMENTS

Reviewer #1 (Remarks to the Author):

The study conducted by Liu and Zhong investigates the extent of epigenetic variation at ncRNAs in *Arabidopsis thaliana*. They have identified numerous significant methylQTLs, the majority of which are located proximal to the locus. These findings suggest the presence of genetic variants at these loci that likely contribute to the observed methylation variation. The researchers have performed correlations between publicly available trait data and methylation levels, revealing numerous associations. However, it is challenging to determine causality and distinguish between genetic and epigenetic factors. Notably, they have identified a methylQTL at mi157 and convincingly demonstrated that targeting this locus with a methyltransferase fused with the SunTag system induces ectopic methylation and an early flowering phenotype.

Overall, this study is timely given the growing interest in ncRNA biology. The subsequent investigation on miRNA157 provides a compelling example of linking natural methylation variation to phenotypic differences, although it would have been preferable to demonstrate this with an unknown flowering regulator. The manuscript could benefit from toning down some of the claims about links to adaptation/plasticity and providing a more comprehensive review of the literature on methylation QTL from the plant community, considering the extensive body of research in this area.

Major Comments:

1. It is worth noting in the introduction that while miRNAs are technically ncRNAs, they have well-established functions compared to the debated roles of other ncRNAs.
2. The study would benefit from toning down the implications of epialleles in adaptation, as there is no supporting data on adaptation presented in this study. Although epialleles may be important for adaptation, further research is required to establish this.
3. Is the flowering phenotype stable when the transgene is segregated away? Testing whether the phenotype persists in the absence of the transgene would determine whether methylation or the SunTag-targeting system is the causal factor.
4. The contribution of Figure 1F is unclear. How is it controlled? Can the same analysis be performed with protein-coding genes for comparison? Otherwise, there is no basis for establishing a background control.
5. The experiments targeting mi157 methylation are interesting, but it should be noted that it is not technically an epiallele until the targeting transgene has been segregated away. Is the methylation stable in the absence of the transgene, along with the flowering phenotype?
6. Consider conducting bisulfite sequencing of the SunTag lines to determine the specificity of the targeted methylation. How much ectopic targeting occurs in these lines? If DMR analysis is performed, how many DMRs are identified genome-wide between control and targeted lines?

Minor Comments:

1. The manuscript would benefit from language editing.
2. Consider revising the title to "Epiallelic Variation of Non-Coding RNAs and Their Phenotypic Consequences."
3. Provide citations for line 56.
4. Provide citations for line 67. Early studies on this topic conducted genome-wide GWAS, not solely on gene bodies. They might have followed up on genes specifically, but the initial GWAS covered DMRs throughout the genome, regardless of location.
5. The first methylQTL mapping study in *Arabidopsis* by Schmitz et al., *Nature*, 2013, should be cited along with others mentioned on line 66.
6. Edit line 91 for clarity.
7. Correct the spelling of "epigenomes" on line 104.
8. While the correlations are valid, it is unknown whether the cause of the correlation is due to

population structure, likely linked to genetic variations associated with ncRNA genes.

9. Line 204 – Be cautious with these conclusions, as statistical power also influences the ability to detect distant methylQTLs. It is still possible that they exist.

10. Line 224 – This hypothesis requires further experimentation for verification. It is currently a plausible hypothesis.

11. In general, there is a lack of discussion regarding the significance of genetic variants associated with methylation variation. The literature on this topic in plants (including TEs, copy number variants, etc.) is extensive. Line 300 attempts to address this, but it lacks citations to many studies that have focused on genetic variants associated with methylation in Arabidopsis and maize.

Reviewer #2 (Remarks to the Author):

In the manuscript under consideration, the authors conducted a thorough study of DNA methylation of Arabidopsis genomes, focusing on ncRNAs. They first presented the profiling of methylation polymorphisms, and then studied the DNA polymorphisms affecting such methylations via GWAS (using a linear mixed model), discovering novel loci other than known methyltransferases. Focusing on flowering time phenotype, they carried out in-depth investigation of the functional roles of methylations in related biological pathways. In particular, miR157a was studied using CRISPR-dCas9 to show de establishment of DNA methylation and its association to early flowering.

The work looks solid and interesting. I don't have main concerns. The only minor point for the authors' consideration is on its GWAS analysis: would you please show a QQ-plot to demonstrate that the population structure is indeed under control?

Reviewer #3 (Remarks to the Author):

This paper analyses epigenetic variation in plants by mining the extensive database of 811 accession of Arabidopsis. Through GWAS they identified methylQTLs outside known DNA methyltransferases or related genes and many of them were located close to non-coding RNA genes. The Methyl QTLs are enriched in specific chromatin states and they could correlate methylation of 1400 ncRNA genes with many plant adaptation phenotypes, including flowering time. Then they tethered a DNA methyltransferase MQ1 to the mir157a locus and showed that this led to reduction in miR157 expression and an early flowering phenotype. They have correlated these changes with reduction in expression of SPL targets. They propose that epigenetic variations in ncRNA genes may contribute to plant diversity and adaptation. The paper is interesting and provide significant correlative information genome-wide about the role of ncRNAs in accession adaptation. However, there are several points that may require further analysis:

1. Fig. 1e there are significant differences between snoRNA, snRNA genes and both long ncRNAs and protein-coding genes. The statement in line 137-138 needs to be clarified.

2. The significant correlation in cytoscape needs to be better presented. What is the difference between development and root morphology to claim that high correlation with plant adaptation phenotypes exist?

3. They have constructed 2 KD miR157 lines and 2 overexpressor lines which showed opposite phenotypes. However, the expression of targets is not correlative (e.g. SPL3 is increased in KD lines but not reduced in OE lines; SPL10 have opposite expression between the 2 lines; even for SPL11,13,15 two OE lines differed?). This needs to be clarified or repeated?

4. Then they constructed 6 transgenic lines with pmiR157:MIR157A constructs. What for? This is partial overexpression (copy number variation?) compared to previous OE lines. It will be good to include the previous OE ones. Furthermore, to conclude that Fig. 3H and Fig. 3i are "correlative" maybe an over-interpretation? Normally lines 5 and 6 should have longer flowering times than the others?

5. They use the methylQV1 strategy to induce DNA methylation on miR157a locus where a minor change in DNA methylation is observed (Fig. 4b). What happens with miR157a expression? Here SPL3 is strongly upregulated compared to other SPLs?

6. The DNA methylation targeting by using fusions of MQ1 to Cas9 was already described but it is very nice that the authors leverage the potential of the approach to knockdown a miR157 gene in this work. As expected, there is a related physiological output, i.e. early flowering. What is missing here is a deeper assessment of what may happen later, comparing the impact of epigenomic variation vs genomic variation. How do the next generations of segregated clean lines behave? Is the epiallele maintained through the next generation at least?

7. When this change in methylation occurs in natural accessions? Is it gradually or suddenly during flowering? Can some links with environmental clues be deduced from the 1st part of the analysis (vernalization, latitude, longitude, cytoscape network?) with miR157 methylQTL? It may be the case but this should be better highlighted.

REVIEWER COMMENTS:

Reviewer #1:

The study conducted by Liu and Zhong investigates the extent of epigenetic variation at ncRNAs in *Arabidopsis thaliana*. They have identified numerous significant methylQTLs, the majority of which are located proximal to the locus. These findings suggest the presence of genetic variants at these loci that likely contribute to the observed methylation variation. The researchers have performed correlations between publicly available trait data and methylation levels, revealing numerous associations. However, it is challenging to determine causality and distinguish between genetic and epigenetic factors. Notably, they have identified a methylQTL at miR157 and convincingly demonstrated that targeting this locus with a methyltransferase fused with the SunTag system induces ectopic methylation and an early flowering phenotype.

Overall, this study is timely given the growing interest in ncRNA biology. The subsequent investigation on miRNA157 provides a compelling example of linking natural methylation variation to phenotypic differences, although it would have been preferable to demonstrate this with an unknown flowering regulator. The manuscript could benefit from toning down some of the claims about links to adaptation/plasticity and providing a more comprehensive review of the literature on methylation QTL from the plant community, considering the extensive body of research in this area.

Response: We appreciate this reviewer's positive comments and helpful suggestions to improve this improvement.

Major Comments:

Q1. It is worth noting in the introduction that while miRNAs are technically ncRNAs, they have well-established functions compared to the debated roles of other ncRNAs.

Response: We appreciate this comment and have added one sentence in the introduction part to capture this point (Lines 85-86).

Q2. The study would benefit from toning down the implications of epialleles in adaptation, as there is no supporting data on adaptation presented in this study. Although epialleles may be important for adaptation, further research is required to establish this.

Response: Thanks for this suggestion. As suggested, we have revised the abstract and main text to tone down the implications of epialleles in adaptation.

Q3. Is the flowering phenotype stable when the transgene is segregated away? Testing whether the phenotype persists in the absence of the transgene would determine whether methylation or the SunTag-targeting system is the causal factor.

Response: We appreciate this comment. As suggested, we obtained the F₂ and F₃ progenies with and without transgene from heterozygous parental lines (Figure R1a). We genotyped and then determined the flowering time of individual F₂ and F₃ progeny. We found that the flowering time of both transgene-free and transgene-containing (either homozygous or heterozygous) plants showed significant earlier flowering phenotype compared with Col-0 in both populations.

Further comparison between transgene-free and transgene-containing plants revealed no significant flowering phenotype difference (Figure R1b). This result confirmed that the early flowering time phenotype is not caused by the transgene insertion and is stable when the transgene is segregated away.

Figure R1. Flowering phenotype is stable when transgene is segregated away.

Q4. The contribution of Figure 1F is unclear. How is it controlled? Can the same analysis be performed with protein-coding genes for comparison? Otherwise, there is no basis for establishing a background control.

Response: We apologize for the confusion. We performed the Person's correlation test between 12,115 methylation data of 4,228 ncRNA genes and 303 phenotypes. As mentioned in the main text, we used a highly stringent threshold ($P < 6.08 \times 10^{-5}$, $N \geq 100$) and only kept significant correlations to construct the network. Most of the correlations were non-significant. Thus, the non-significant correlations serve as the background control. The same analysis could also be performed with protein-coding genes. However, we feel that the non-significant ncRNAs is a better background control comparing with protein-coding genes. For clarity, we have revised the Fig. 1f and edited the legend with the following sentences:

“The network of significant correlations between methylation of ncRNA genes and plant phenotypes. Each dot indicates one ncRNA gene and each line indicates significant correlation between the DNA methylation of this gene and the plant phenotype.”

Q5. The experiments targeting mi157 methylation are interesting, but it should be noted that it is not technically an epiallele until the targeting transgene has been segregated away. Is the methylation stable in the absence of the transgene, along with the flowering phenotype?

Response: As suggested, we have performed quantitative PCR following McrBC digestion of gDNA extracted from multiple randomly selected individuals from F₂ population (see Figure R1). As shown in Figure R2, we observed increase of DNA methylation in three transgene-containing and six transgene-free plants comparing with Col-0. This result indicated that the increased methylation could be inherited to next generation in the absence of transgene.

Figure R2. Quantitative PCR following McrBC digestion (McrBC-qPCR)-based DNA methylation assay of miR157a

Q6. Consider conducting bisulfite sequencing of the SunTag lines to determine the specificity of the targeted methylation. How much ectopic targeting occurs in these lines? If DMR analysis is performed, how many DMRs are identified genome-wide between control and targeted lines?
Response: As suggested, we have performed whole-genome bisulfite sequencing of SunTag-MQ1v line and determined its DMRs against WT control using threshold of 40% CG difference in 100bp windows. Totally, we identified 81 hyper- and 33 hypo-CG DMRs (Table R1). We also found some CHG and CHH DMRs. As a comparison, we re-analyzed published WGBS data of SunTag-MQ1v lines from the Jacobsen lab (Ghoshal et al., 2021) using same pipeline and their own control. Our data showed less CG and CHG DMRs but variations in the numbers of CHH DMRs. We don't know whether these CHH DMRs are real or function. The non-CG DMRs might be due to the nature of non-CG methylation plasticity. Similar large non-CG variations were observed among 54 high-quality WGBS data of Col-0 from worldwide (Zhang et al., 2018).

Table R1. DMR analysis and comparison

Genotype	CG		CHG		CHH	
	Hyper	Hypo	Hyper	Hypo	Hyper	Hypo
MQ1v vs Col-0 (This study)	81	33	4,713	1,377	44,834	9,944
FWAg4_SunTag22aa T2_1minus (PNAS)	1,674	169	8,848	1,275	28,008	5,167
FWAg4_SunTag22aa T2_1plus (PNAS)	1,850	143	9,765	1,155	30,671	4,639
FWAg4_SunTag22aa T2_2minus (PNAS)	920	546	5,253	4,789	15,988	17,310
FWAg4_SunTag22aa T2_2plus (PNAS)	1,513	296	7,812	1,917	23,938	6,203

Reference for this response:

Ghoshal B, Picard CL, Vong B, Feng S, Jacobsen SE (2021) CRISPR-based targeting of DNA methylation in *Arabidopsis thaliana* by a bacterial CG-specific DNA methyltransferase. *Proc Natl Acad Sci U S A* 118(23): e2125016118.

Zhang Y, Harris CJ, Liu Q, Liu W, Ausin I, Long Y, Xiao L, Feng L, Chen X, Xie Y, Chen X, Zhan L, Feng S, Li JJ, Wang H, Zhai J, Jacobsen SE (2018) Large-scale comparative epigenomics reveals hierarchical regulation of non-CG methylation in *Arabidopsis*. *Proc Natl Acad Sci U S A* 115(5): E1069-E1074.

Minor Comments:

Q7. The manuscript would benefit from language editing.

Response: Thanks for this suggestion. We have sought the help from our departmental science writer for language editing. We have significantly improved the clarity in this revised manuscript.

Q8. Consider revising the title to "Epiallelic Variation of Non-Coding RNAs and Their Phenotypic Consequences."

Response: We appreciate this suggestion. As suggested, we have changed the title accordingly.

Q9. Provide citations for line 56.

Response: As suggested, we have added citations for this sentence.

Q10. Provide citations for line 67. Early studies on this topic conducted genome wide GWAS, not solely on gene bodies. They might have followed up on genes specifically, but the initial GWAS covered DMRs throughout the genome, regardless of location.

Response: As suggested, we have added proper citations. We have also edited this sentence to include TEs and DMR regions for accuracy as following:

"Current epiallele studies have mostly focused on the protein-coding genes, transposable elements (TEs), and differentially methylated regions on the genome^{8,13-15}. Little is known about the epialleles of ncRNA genes." (Lines 67-69).

Q11. The first methylQTL mapping study in *Arabidopsis* by Schmitz et al., Nature, 2013, should be cited along with others mentioned on line 66.

Response: We apologize for this unintentional omission. We have cited this study on line 66. We have also added this citation in the following sentence for DMR GWAS (Line 68).

Q12. Edit line 91 for clarity.

Response: We have revised this sentence to be "In natural accession populations, both genetic and epigenetic changes may contribute to the variation of complex diseases and

phenotypes. Thus, it is challenging to dissect the epigenetic function from genetic contribution.” (Lines 92-94).

Q13. Correct the spelling of "epigenomes" on line 104.

Response: We have corrected the spelling (Line 106). Thanks.

Q14. While the correlations are valid, it is unknown whether the cause of the correlation is due to population structure, likely linked to genetic variations associated with ncRNA genes.

Response: We agree with this point. As our response to Q12, it is challenging to separate the effect of epigenetic change from the tightly linked genetic variations or population structure solely based on statistical analysis. This is why we applied a highly stringent threshold for these significant correlations ($P < 6.08 \times 10^{-5}$, $N \geq 100$) in hoping to minimize false positive rate. However, confirming these correlations is very challenging.

Q15. Line 204 – Be cautious with these conclusions, as statistical power also influences the ability to detect distant methylQTLs. It is still possible that they exist.

Response: We agree and appreciate this comment. We have revised this sentence to be: “These results demonstrated the enrichment and the higher PVE of proximal genetic variants in regulating DNA methylation variations of ncRNA genes.” (Lines 208-209).

Q16. Line 224 – This hypothesis requires further experimentation for verification. It is currently a plausible hypothesis.

Response: We agree and have revised this sentence as following:

“Together, these results showed that methylation variations of ncRNA genes were highly correlated with the flowering and adaptation related phenotypes.” (Lines 228-229).

Q17. In general, there is a lack of discussion regarding the significance of genetic variants associated with methylation variation. The literature on this topic in plants (including TEs, copy number variants, etc.) is extensive. Line 300 attempts to address this, but it lacks citations to many studies that have focused on genetic variants associated with methylation in Arabidopsis and maize.

Response: We appreciate these comments. As suggested, we have added the discussion on the significance of other types of genetic variants including TE, repeat sequences, gene duplications (Lines 306-316).

Reviewer #2:

In the manuscript under consideration, the authors conducted a thorough study of DNA methylation of Arabidopsis genomes, focusing on ncRNAs. They first presented the profiling of methylation polymorphisms, and then studied the DNA polymorphisms affecting such methylations via GWAS (using a linear mixed model), discovering novel loci other than known

methyltransferases. Focusing on flowering time phenotype, they carried out in-depth investigation of the functional roles of methylations in related biological pathways. In particular, miR157a was studied using CRISPR-dCas9 to show de establishment of DNA methylation and its association to early flowering. The work looks solid and interesting. I don't have main concerns.

Response: We appreciate the enthusiastic and positive comments from this reviewer on this manuscript.

Q18. The only minor point for the authors' consideration is on its GWAS analysis: would you please show a QQ-plot to demonstrate that the population structure is indeed under control?

Response: Thanks for this good suggestion. We have plotted QQ-plot from the P -values of GWAS result of miR157a CHG methylation and added it in Fig. 3a (also attached below). From the QQ-plot, we confirmed that the population structure was well controlled.

Fig. 3a. QQ-plot for CHG methylation of miR157a

Reviewer #3:

This paper analyses epigenetic variation in plants by mining the extensive database of 811 accession of Arabidopsis. Through GWAS they identified methylQTLs outside known DNA methyltransferases or related genes and many of them were located close to non-coding RNA genes. The Methyl QTLs are enriched in specific chromatin states and they could correlate methylation of 1400 ncRNA genes with many plant adaptation phenotypes, including flowering time. Then they tethered a DNA methyltransferase MQ1 to the mir157a locus and showed that this led to reduction in miR157 expression and an early flowering phenotype. They have correlated these changes with reduction in expression of SPL targets. They propose that epigenetic variations in ncRNA genes may contribute to plant diversity and adaptation. The paper is interesting and provide significant correlative information genome-wide about the role of ncRNAs in accession adaptation.

Response: We appreciate this reviewer's enthusiastic and positive comments.

Q19. Fig. 1e there are significant differences between snoRNA, snRNA genes and both long ncRNAs and protein-coding genes. The statement in line 137-138 needs to be clarified.

Response: We appreciate this comment and have revised this sentence as following:

“In non-CG context, most groups had higher methylation level than protein-coding genes. While in CG context, both sense lncRNA and antisense lncRNA genes had slightly lower DNA methylation comparing with protein-coding genes (protein-coding genes: 16.57 ± 0.02 , lncRNA genes: 16.05 ± 0.03 , antisense lncRNA genes: 12.28 ± 0.02 , Mann-Whitney test $P < 2.00 \times 10^{-15}$, Fig. 1e). This observation explained the global higher CG methylation of protein-coding genes than that of ncRNA genes (Fig. 1a). Notably, miRNA genes showed much higher methylation levels in both CG and non-CG context (Fig. 1e)” (Lines 139-145).

Q20. The significant correlation in cytoscape needs to be better presented. What is the difference between development and root morphology to claim that high correlation with plant adaptation phenotypes exist?

Response: As suggested, we have revised the Fig. 1f (also attached below) for clarity. We did not compare development and root morphology. Previously, we included flowering, development, and root morphology as adaptation related phenotypes. Comparing with ion- and metabolite-related phenotypes, we observed much more significant correlations with adaptation related phenotypes. Following suggestion from Reviewer#1 (Q2), we toned down our conclusion on the implications of epialleles in adaptation and only focused on the phenotypes. We have revised this sentence to be: “we found that most (87%, 1,801 of 2,072) significant correlations fell into groups of fitness, flowering, plant structure, development, and root morphology (Fig. 1f). While for ion-, yield-, and metabolite-related phenotypes, far fewer correlations (13%, 271 of 2,072) were detected (Fig. 1f).” (Lines 157-160).

Fig. 1f. The network of methylation of ncRNA genes and plant phenotypes

Q21. They have constructed 2 KD miR157 lines and 2 overexpressor lines which showed opposite phenotypes. However, the expression of targets is not correlative (e.g. SPL3 is increased in KD lines but not reduced in OE lines; SPL10 have opposite expression between the 2 lines; even for SPL11,13,15 two OE lines differed?). This needs to be clarified or repeated?
Response: We appreciate this comment. As suggested, we have performed extra biological repeats and found reproducibly significant expression changes of all tested *SPL* genes except for *SPL13* (Fig. 3f, g). Thus, we did not include the *SPL13* in this revised manuscript.

Fig. 3f, g. The expression changes of *SPL* genes in miR157a knockdown and overexpression transgenic lines

Q22. Then they constructed 6 transgenic lines with pmiR157:MIR157A constructs. What for? This is partial overexpression (copy number variation?) compared to previous OE lines. It will be good to include the previous OE ones. Furthermore, to conclude that Fig. 3H and Fig. 3i are “correlative” maybe an over-interpretation? Normally lines 5 and 6 should have longer flowering times than the others?

Response: We apologize for the confusion. Here, we aim to investigate the relationship of miR157a abundance with flowering phenotype. However, the individual 35S OE lines showed similar high expression levels. Thus, we generated the endogenous promoter-driven miR157a expression lines (pmiR157:MIR157A) to closely mimic the spatial-temporal expression pattern as native miR157a. As shown in Supplementary Fig. 6 (also attached below), we observed a

strong correlation (Pearson's $r = 0.8$; $N=6$). Indeed, we did observe longer flowering times of line 5 and line 6 comparing with other lines.

Supplementary Fig. 6. Pearson's correlation between abundance of miR157a and flowering time in transgenic lines driving with native promoter

Q23. They use the methylQV1 strategy to induce DNA methylation on miR157a locus where a minor change in DNA methylation is observed (Fig. 4b). What happens with miR157a expression? Here SPL3 is strongly upregulated compared to other SPLs?

Response: As described in main text (Lines 273-274) and Fig. 4f, the abundance of mature miR157a was significantly decreased in SunTag-MQ1v comparing with WT. We observed that SPL3 was strongly upregulated compared to other SPLs consisting with other studies (He et al., 2018; Jung et al., 2011). This is likely due to the fact that SPL3 is co-regulated by miR156/157 and miR172, whose expression is promoted by miR156/157 (Jung et al., 2011; Wu et al., 2009).

References for this response:

He J, Xu M, Willmann MR, McCormick K, Hu T, Yang L, Starker CG, Voytas DF, Meyers BC, Poethig RS (2018) Threshold-dependent repression of *SPL* gene expression by miR156/miR157 controls vegetative phase change in *Arabidopsis thaliana*. *PLoS Genet* 14: e1007337

Jung JH, Seo PJ, Kang SK, Park CM (2011) miR172 signals are incorporated into the miR156 signaling pathway at the *SPL3/4/5* genes in *Arabidopsis* developmental transitions. *Plant Mol Biol* 76: 35-45

Wu G, Park MY, Conway SR, Wang JW, Weigel D, Poethig RS (2009) The sequential action of miR156 and miR172 regulates developmental timing in *Arabidopsis*. *Cell* 138: 750-759

Q24. The DNA methylation targeting by using fusions of MQ1 to Cas9 was already described but it is very nice that the authors leverage the potential of the approach to knockdown a miR157 gene in this work. As expected, there is a related physiological output, i.e. early flowering. What is missing here is a deeper assessment of what may happen later, comparing the impact of epigenomic variation vs genomic variation. How do the next generations of segregated clean lines behave? Is the epiallele maintained through the next generation at least?

Response: Please see our responses to Q3 and Q5 from reviewer #1(see above). We observed the early flowering phenotype and increased CG methylation of miR157a in the next generation progenies without the transgene. This indicated that this epiallele and the phenotype is inheritable for at least one more generation.

Q25. When this change in methylation occurs in natural accessions? Is it gradually or suddenly during flowering? Can some links with environmental clues be deduced from the 1st part of the analysis (vernalization, latitude, longitude, cytoscape network?) with miR157 methylQTL? It may be the case but this should be better highlighted.

Response: Thanks for these interesting questions. As mentioned in main text (Lines 235-236), we found a significant positive correlation between DNA methylation of miR157a and longitude (Pearson's $r = 0.15$, $N = 768$, $P = 3.51 \times 10^{-5}$). However, it is currently challenging to trace the methylation changes of miR157a in different generations and environments due to the lack of necessary data, such as the abundance data of miR157a in natural accessions. The "epimutation-clock" could be used to estimate the divergence of DNA methylation of miR157a in natural accessions. While multiple tools to estimate epigenetic age in animals are available, such tool for this purpose in *Arabidopsis* is still lacking. Recently, Yao et al. (2023) reported the application of epigenetic clock in *Arabidopsis*, but the details of the method/software have not yet been made available.

Reference for this response:

Yao N, Zhang Z, Yu L, Hazarika R, Yu C, Jang H, Smith LM, Ton J, Liu L, Stachowicz J, Reusch T, Schmitz RJ, Johannes F (2023) An evolutionary epigenetic clock in plants. bioRxiv [Preprint]. doi: 10.1101/2023.03.15.532766.

REVIEWER COMMENTS

Reviewer #1 (Remarks to the Author):

Thank you for addressing my comments so thoroughly. This will make a nice contribution to the field.

Reviewer #2 (Remarks to the Author):

From the QQ-plot, one can see that the P-value curve deviates from the diagonal from $-\log(P) < 2$. So, it appears that the population structure is not under control. The authors may consider revisiting the procedure to adjust this part. Alternatively, if the authors and editor find the biological validation more important (in contrast of the statistical process being rigorous), I am fine if they just publish as is.

Reviewer #3 (Remarks to the Author):

The paper has been significantly improved and all my comments have been addressed in a satisfactory way. The inheritance of the epigenetic changes induced in the miR157 locus in the absence of the transgene clearly add a major novelty to the paper. The other correlations and changes in gene expression (notably SPL targets) are now much better presented to support the global statements. In addition, the down-sizing the conclusions on adaptation phenotypes (from the network analysis) allow them to present a more careful evaluation of their results.

REVIEWER COMMENTS:

Reviewer #1:

Thank you for addressing my comments so thoroughly. This will make a nice contribution to the field.

Response: We are happy that this reviewer was satisfied with our revision.

Reviewer #2:

From the QQ-plot, one can see that the P-value curve derives from the diagonal from $-\log(P) < 2$. So, it appears that the population structure is not under control. The authors may consider revisiting the procedure to adjust this part. Alternatively, if the authors and editor find the biological validation more important (in contrast of the statistical process being rigorous), I am fine if they just publish as is.

Response: We appreciate this comment. As a further examination whether the population structure is under the control, we have performed QQ-plots of additional 50 GWAS results with methylation QTL. Three QQ-plots showed that the P-value curve derives from the diagonal from $-\log(P)$ close to 2 (one representative example is shown in Figure R1B). The remaining 94% (47 out of 50) of GWAS with methylation QTL results showed a P-value curve deriving from the diagonal from $-\log(P) > 2$ (six representative examples are shown in Figure R1C). This extensive analysis suggests that the population structure is well controlled. We further checked multiple published GWAS studies, including human, rice, maize, and *Arabidopsis*, and found earlier or similar separations of expected and observed P-values comparing with our results ($-\log(P)$ close to 2 in Figure 3a; representative examples shown in Figure R2). Thus, we are confident with our statistical analysis.

Figure R1. Examples of quantile-quantile plots

A, One representative QQ-plot of GWAS results without methylation QTL. **B**, One of three QQ-plots of GWAS results with methylation QTL and the P-value curve derives from the diagonal from $-\log(P)$ close to 2. **C**, Six representative QQ-plots of GWAS results with methylation QTL and the P-value curves derive from the diagonal from $-\log(P) > 2$.

[redacted]

Figure R2. Examples of quantile-quantile plots from other published studies

References for this response:

Nagel, M. et al. Meta-analysis of genome-wide association studies for neuroticism in 449,484 individuals identifies novel genetic loci and pathways. *Nat Genet* 50, 920-927 (2018).

International League Against Epilepsy Consortium on Complex Epilepsies. GWAS meta-analysis of over 29,000 people with epilepsy identifies 26 risk loci and subtype-specific genetic architecture. *Nat Genet* 55, 1471-1482 (2023).

Atwell, S. et al. Genome-wide association study of 107 phenotypes in *Arabidopsis thaliana* inbred lines. *Nature* 465, 627-31 (2010).

Huang, X. et al. Genome-wide association studies of 14 agronomic traits in rice landraces. *Nat Genet* 42, 961-7 (2010).

Reviewer #3:

The paper has been significantly improved and all my comments have been addressed in a satisfactory way. The inheritance of the epigenetic changes induced in the miR157 locus in the absence of the transgene clearly add a major novelty to the paper. The other correlations and changes in gene expression (notably SPL targets) are now much better presented to support the global statements. IN addition, the down-sizing the conclusions on adaptation phenotypes (from the network analysis) allow them to present a more careful evaluation of their results.

Response: We appreciate this reviewer's enthusiastic and positive comments.

REVIEWERS' COMMENTS

Reviewer #2 (Remarks to the Author):

My comment has been fully addressed.